# Stakeholder Perceptions Can Distinguish 'Paper Parks' from Marine Protected Areas

**Veronica Relano** [1,*] , **Tiffany Mak** [1] , **Shelumiel Ortiz** [2] **and Daniel Pauly** [1]

1  Sea Around Us, Institute for the Oceans and Fisheries, University of British Columbia, Vancouver, BC V6T 1Z4, Canada
2  Quantitative Aquatics, Khush Hall, IRRI (International Rice Research Institute), Los Baños 4031, Philippines
*  Correspondence: v.relano@oceans.ubc.ca

**Abstract:** While numerous Marine Protected Areas (MPA) have been created in the last decades, their effectiveness must be assessed in the context of the country's biodiversity conservation policies and must be verified by local observations. Currently, the observations of local stakeholders, such as those from non-governmental organizations (NGOs), academics, government civil servants, journalists, and fishers, are not considered in any MPA database. The Sea Around Us has added observations from local stakeholders to address this gap, adding their perspectives to its reconstructed fisheries catch database, and to at least one MPA in each country's Exclusive Economic Zone. It is important to pursue and incentivize stakeholder knowledge sharing to achieve a better understanding of the current level of marine protection, as this information is a valuable addition to the existing MPA databases. To address this gap, we demonstrated that personal emails containing a one-question questionnaire about the fishing levels in an MPA are an excellent way to gather data from local stakeholders, and that this works especially well for respondents in NGOs, academia, and governments. Of the stakeholders who replied to our personalized email, 66% provided us with the fishing level of the MPA that we asked for. The paper also presents how to access this information through the Sea Around Us website, which details in anonymized form the most common fishing levels for each selected MPA, as perceived or observed by different local stakeholder groups. This information is a unique and novel addition to a website that is concerned with marine conservation and contributes to a more accurate and inclusive discourse around MPAs. This information also helps to identify the gaps that need to be addressed to turn 'paper parks' (i.e., MPAs that are legally designated but not effective) into effective MPAs, which can contribute to climate-resilient 'blue economies'.

**Keywords:** marine conservation; biodiversity; fishing pressure; effective protection; stakeholders' observation; local participation; questionnaire

## 1. Introduction

Marine Protected Areas (MPAs) are bodies of water that are mainly managed for biodiversity conservation purposes [1]. MPAs can be used as a tool to protect marine life [2], and can contribute to the socio-economic development of an area while supporting traditional cultures. In 2010, during a meeting of the Convention on Biological Diversity, many countries around the world agreed to protect at least 10% of the marine water bodies under their jurisdiction by 2020 [3,4]. This target was not met, with only 7.7% of the oceans currently protected [5]. The new target is to protect 30% of the world's land and water surfaces by 2030 [6]. Research has shown that focusing on implementing large MPAs over many small MPAs is a more cost-effective way to move forward in the protection of the oceans [7–10].

The Sea Around Us is a research initiative at the University of British Columbia which established the first global database of MPA. This was accomplished mainly by separating

the marine from the terrestrial components of the protected spaces in the World Database on Protected Areas [11], and adding to the resulting database, which was called "MPA Global." MPA Global was abandoned, however, when the more comprehensive 'Marine Protection Atlas' (MPAtlas) of the Marine Conservation Institute (https://marine-conservation.org/; accessed 20 April 2022) became available. This database and website (www.mpatlas.org; accessed 1 April 2022) began assessing the regulations within selected MPAs, including their allowed and prohibited activities, at the end of 2021 (e.g., mining, dredging, anchoring, extraction, development, fishing, aquaculture).

Based on the data submitted annually by its member countries, the Food and Agriculture Organization of the United Nations (FAO) createda global database of annual fisheries landings (catches minus discards), with marine fisheries landings assigned to 19 very large 'FAO Statistical Areas' [12]. The Sea Around Us improved these data by adding estimates of the discards and likely illicit catches, which were not considered in the FAO database [13,14]. The Sea Around Us also added catch estimates for fisheries which are often under- or unreported (i.e., artisanal fisheries) [15], or are almost always completely ignored (i.e., subsistence and recreational fisheries) [16,17]. This addition resulted in annual global marine fisheries catch data (from 1950 on), which was assigned to 150,000 ice-free $\frac{1}{2}$ degree latitude and longitude spatial cells [18]. Based on this extensive catch database, the Sea Around Us also provides catch-derived indicators of the status of fisheries by the categories of country, large marine ecosystems, and other geographies [19,20].

The new product from the Sea Around Us described below establishes a bridge between these detailed fisheries data and many of the MPAs included in the MPAtlas. By combining fishery data with MPA information from the literature and stakeholder observations, the new product presented here provides a more comprehensive understanding of the conservation status of our oceans, and should allow for a better assessment of what is protected *de facto*, i.e., in reality, or actual, as opposed to theoretical or aspirational. To the best of our knowledge, no other MPA database or research has assessed or published the observations and perceptions that local stakeholders have about their MPA's protection status, offering different perspectives and information to assess how much and how well our oceans are protected. This new feature, which was recently added to the Sea Around Us website, is a starting point to establish a better understanding of the *de facto* protection and conservation status of MPAs globally. This is important when working towards conservation targets, as relying on official data limits our ability to accurately inform and drive socio-ecological changes on the ground.

This paper presents how the information was collected into a multi-attribute database with observations from local stakeholders, which helps to demonstrate the actual level of protection of a representative set of MPAs on a global scale. We also demonstrate that a one-question questionnaire, sent via personalized emails to stakeholders from all over the world, is a valid and easily replicable method to obtain a better understanding of the status of Marine Protected Areas.

This paper examines on a global scale the response rate of stakeholders, who were asked about their perception of the fishing intensity within the MPA that they were familiar with. We highlight the importance of considering local stakeholders equally and collectively when assessing an MPAs' protection status, in this case via a standard one-question questionnaire. In doing so, we seek to uncover how to make it possible to consider the views of stakeholder groups that are associated with marine conservation within a given country, and how to help distinguish 'paper parks' from *de facto* Marine Protected Areas.

## 2. Materials and Methods

The Sea Around Us' 'EEZ pages' (Exclusive Economic Zones), which present fisheries data for each maritime country of the world, were modified to include a link to our new website feature: 'Marine Protected Areas,' which leads to a page with a brief narrative on the particular country's effort to protect its marine biodiversity, notably in the form of treaties or conventions that it is a member of (Figure 1). For large countries with several

EEZ 'chunks' (Brazil, France, Russia, UK, US, etc.), only one of the EEZ chunks will include detailed information on an MPA. However, all EEZ chunks lead to information on the marine biodiversity protection offered by the country in question, and provide links to both the EEZ chunk with the studied MPA and to the MPAtlas' coverage of that country.

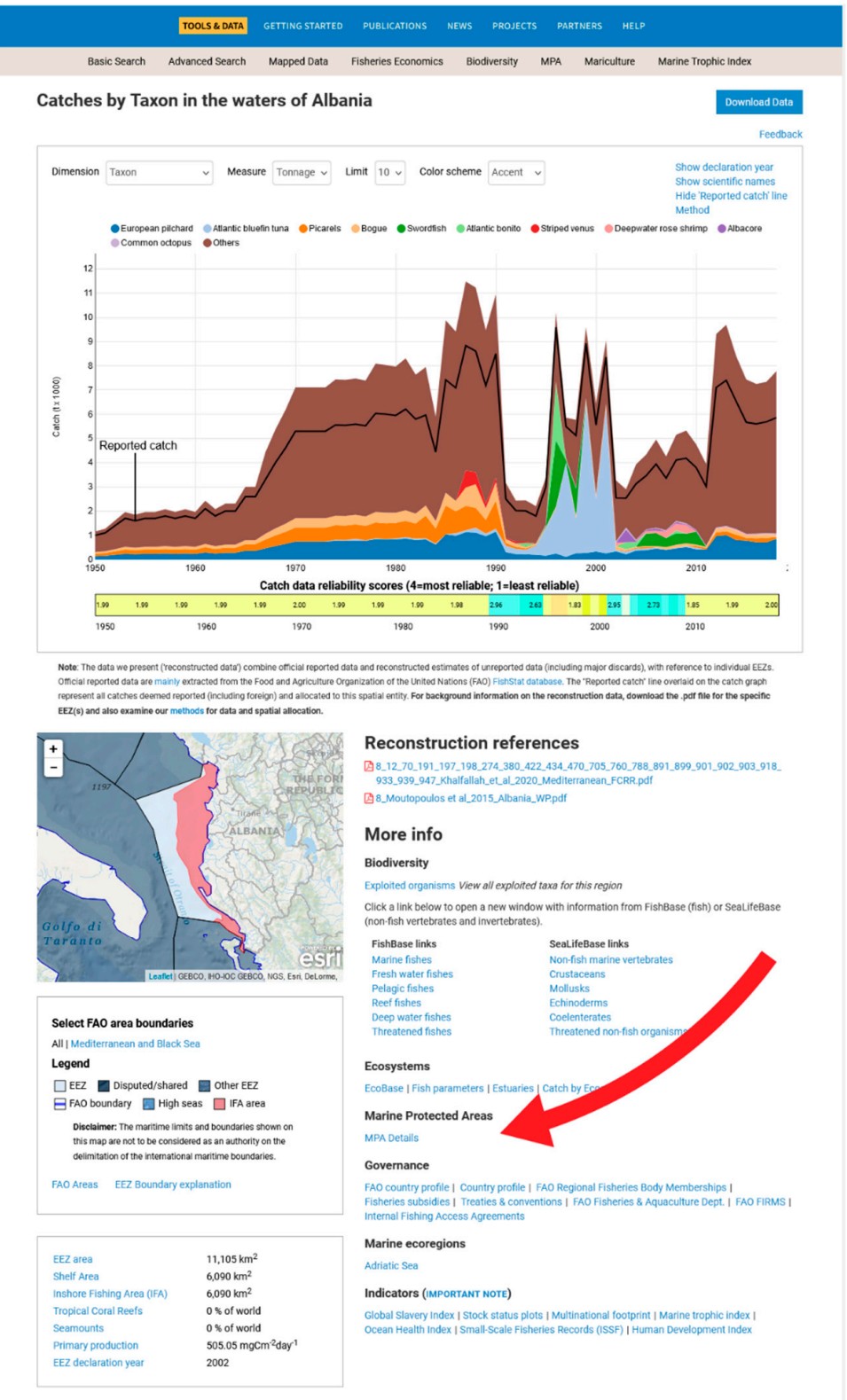

**Figure 1.** The country-level data site allows users to access Marine Protected Areas' details under the 'more info' section.

For most countries (excluding their overseas territories), one major MPA is presented in terms of its area, official designation, and effectiveness, as assessed in the published literature and via the questionnaire sent to the local stakeholders (Figure 2). The bulk of this information was adapted from the marine biodiversity protection sections of the country-by-country catch reconstruction updates, in the reports edited by Derrick et al. [21,22].

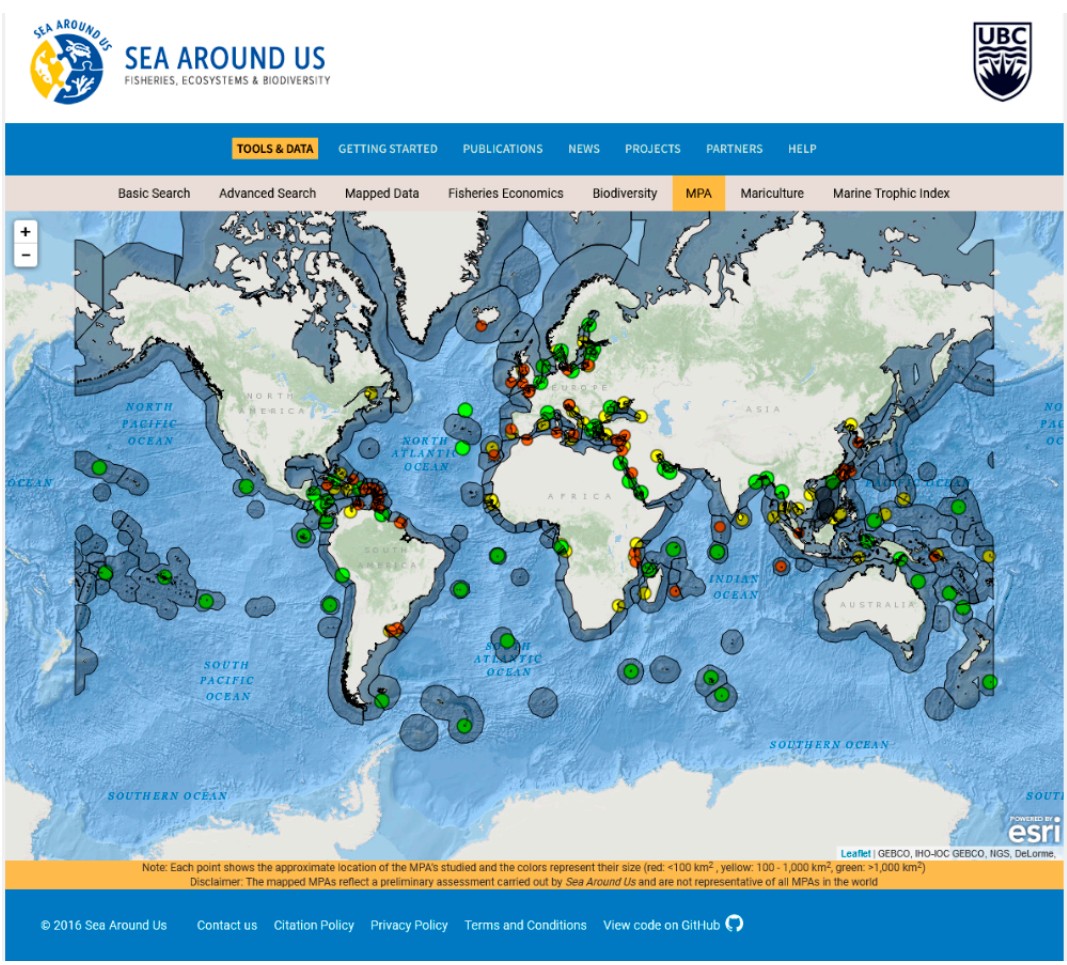

**Figure 2.** A world map of the Marine Protected Areas studied by the Sea Around Us can be accessed via the 'Tools & Data' menu. Each point shows the approximate location of the MPAs studied and the colours represent their size (red: <100 km$^2$, yellow: 100–1000 km$^2$, green: >1000 km$^2$).

When clicking on the MPA details, accessed from the EEZ page of a country under the 'more info' section (see the arrow in Figure 1), or from the global MPA map (see the dots in Figure 2), the MPA webpages have the following elements (from top-down and from left-right): a small map displaying the location and size of the MPA within the EEZ; a brief review of the marine biodiversity protection status of the country; a table with the most frequent fishing level; quotes; and references (Figure 3). On the map, the dots were used to indicate the location of the MPAs, with three different colours to represent different size ranges. Green dots were used for the large MPAs (greater than 1000 km$^2$), yellow dots for medium MPAs (from 100 km$^2$ to 1000 km$^2$), and red dots for small MPAs (less than 100 km$^2$).

From March 2020 to October 2021, we sent 3027 introductory emails to potential stakeholders, i.e., academics, staff of non-governmental organizations (NGOs), journalists, civil servants, and fisher representatives; these people appeared to have close relationships with various MPAs due to their expertise, profession, or network. For each specific MPA, at least one email per stakeholder group was sent to have a representation of each group of stakeholders. Because some of the recipients in this dataset were not randomly chosen and

thus received differing levels of sampling effort, a response rate analysis was not possible, as the probability of receiving a response is dependent on the sampling effort. In the current paper, we present therefore only a description of the sample.

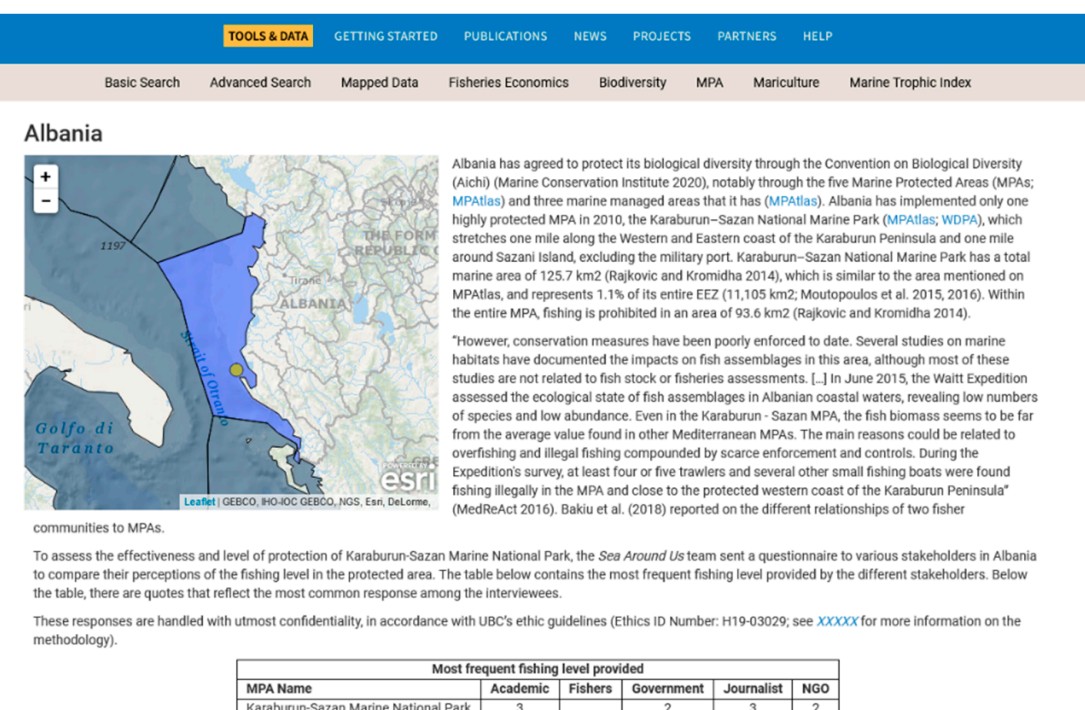

**Figure 3.** The country-level MPA page also shows a map with the MPA's approximate location and biodiversity information details of the country. It is possible to either access this page from the global MPA map (the dots in Figure 1) or the EEZ page of a country under the 'more info' section (highlighted with an arrow in Figure 2).

The emails were personalized (i.e., they did not include a form or an automated form filling) and consisted of only one multiple-choice question on what they thought was the fishing intensity in a no-take MPA in their country. That is, we asked them to provide their perspective on the largest no-take MPA, or (in the absence of a fully no-take MPA), a partially no-take MPA, or the largest MPA if no-take MPAs of any kind did not exist in their country. For the question, we offered the following possible answers: 'no fishing', 'light fishing', 'moderate fishing', and 'very intense fishing'. However, these terms were deliberately left undefined.

The email had a personalized salutation and a text approved by UBC's Office of Research Ethics for studies involving humans (# H19-03029, 29 November 2019). All emails included the name of the first author (V.R.) as the sender and a copy email was sent to the last author (D.P.). It mentioned the purpose of the email, the one-question questionnaire, and mentioned that additional information was welcome, along with the details of the research ethics considered.

The responses of the emailed stakeholders are presented on the Sea Around Us website, in the form of a table such as that of Figure 3, which lists, for each MPA covered, the most commonly perceived level of actual protection by different stakeholders.

For the analysis, in the case of ties, i.e., where multiple fishing levels had the same frequency for the same stakeholder group, the following rules were applied to avoid assuming higher intensity levels in the presence of contradictory information:

(1) When two adjacent fishing levels (e.g., 2 and 3, or 3 and 4) were suggested, that which was better documented (i.e., provided with the best additional information) was selected. When both were provided without additional information, the lower fishing level was selected;

(2) In the case of a tie between fishing levels separated by one level (i.e., 1 and 3, or 2 and 4), the intermediate level was selected;

(3) In the case of a tie between fishing levels 1 and 4, the fishing level selected was 2.

(4) In the case of a tie between fishing levels 1, 2, 3 and 4, we conservatively selected fishing level 2.

Anonymized quotes from our exchanges with stakeholders are also presented on the Sea Around Us MPAs page site (Figure 3) to illustrate the thought process behind the common fishing level responses. The introductory narrative, fishing level responses, quotes, and references were added to the Sea Around Us' database system as CSV files, to allow for easy access and updates. Moreover, while inputting the information into the database and adequately presenting the information on the website, tags were added to words that needed to be hyperlinked, italicized, underlined, and/or bolded. Most of the MPAs were hyperlinked to MPAtlas and Protected Planet to provide the most up-to-date information, allowing users to expand their knowledge by travelling between those websites and ours. Fish species that are mentioned on the webpage were linked to FishBase (www.fishbase.org; accessed on 15 March 2022) and other marine animals to SeaLifeBase (www.sealifebase.org; accessed on 15 March 2022) through hyperlinks, allowing users to find more information about the species in question, such as the distribution range, size, life cycle, threat status, etc.

## 3. Results and Discussion

The new MPA pages of the Sea Around Us website went online on 9 March 2022. As mentioned above, they can be either accessed via the EEZ page of a country (Figure 1) or the 'Tools & Data' tab on the toolbar (Figure 2). In the case of access via the 'Tools & Data' tab, a global map will appear with the MPAs that we studied as dots. These dots, presented in Figure 2, are a new feature of the Sea Around Us database, which provides an idea of the wide geographic reach of our study. When users hover over and click a dot, they are led to details concerning the account of biodiversity protection offered by the country in the EEZ where the MPA is located, and to the account on that specific MPA (Figure 3).

In response to the 3027 emails sent to different stakeholders, we received 1241 responses (a 41% response rate), of which 814 answered the one-question questionnaire about the fishing intensity within the MPA. The 1241 responses included 450 from academia, 337 from civil servants, 298 from NGOs, 83 from journalists, and 73 from fishers. The 814 respondents that answered the one-question-questionnaire were made up of 303 academics, 228 civil servants, 199 NGOs, 44 fishers, and 40 journalists (Figure 4). The results in Figure 4 suggest that different stakeholders differ sharply in their approachability and relationship to MPA governance and status. The lack of effective participation is mentioned in the literature, particularly where top-down governance is practiced; conversely, the engagement of the community improves MPA performance [23].

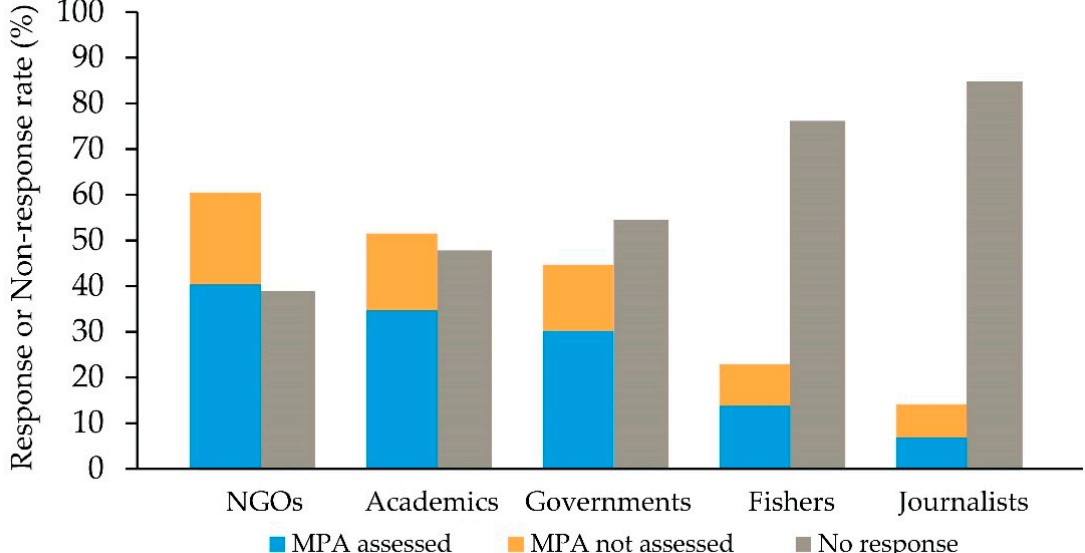

**Figure 4.** Response and non-response rates (in percentages) to our one-question questionnaire by stakeholder groups. The response 'MPA assessed' indicates that the recipient responded to at least one email with a fishing level, while 'MPA not assessed' means that the recipient responded to at least one email but did not provide a fishing level category. 'No response' means that the recipient did not respond to any email they received.

More granular analysis of the data received from fishers revealed a similar response rate among the types of fishers (29% recreational, 25% industrial, and 20% artisanal). The number of fishers that replied with an answer about the fishing intensity level within the MPA was also similar among the three categories (15% artisanal, 15% industrial and 14% recreational). In many cases, fishers do not perceive any benefits in the short-term concerning MPAs and often resist their creation [23]. Through our research, we saw that the participation was not very high, but the artisanal fishers were willing to reply with a fishing level, despite being the group most likely to be adversely impacted in the short-term by MPAs [23,24]. The participation of artisanal fishers, despite being relatively low in this study, is of vital importance because of their close relationship with the sea, which makes them aware of how the environment relates to the psychological, economic, and social aspects of their life [25,26]. This reinforces the important role of considering the ecological knowledge of fishers, as they can provide a solid basis for MPA design and implementation [27] by providing unique information about the habitat, behaviour, abundance, and historical changes of targeted species [28]. Additionally, fishers can provide key insights into the inshore fishing effort in developed and developing countries, particularly in the absence of data from the Vessel Monitoring System [29]. They are also among the first to observe and experience negative or positive impacts [30]. Alexander et al. (2018) [31] suggested that when local fishers are considered in conservation plans and have the chance

to participate in the governance of MPAs, they are more likely to report illegal fishing; even the co-management of MPAs becomes feasible [32].

Going forward, we suggest more follow-ups with the local fishers to improve participation using other communication channels, which we avoided in the current research, as we wanted to maintain the same contact methods as those used for the other stakeholder groups. A follow-up using other communication channels, such as social media, also would have been very beneficial in improving the low response rate that we encountered from journalists. We propose that this low rate could be a result of a high number of requests and a low number of journalists in some coastal and especially developing countries. Moreover, most journalists cover a wide variety of topics, but this does not make them knowledgeable about the fishing status of a specific MPA; thus, we assumed that they opted not to reply to the email. We also propose that, in general, the individuals in the journalist group were more used to gathering information themselves than sharing it with other sectors. At the individual MPA level, a low response rate, i.e., the unwillingness to respond, could reflect a lack of awareness of the MPA, lack of support, or other factors that need further research. Overall, the participation of all stakeholders generated a large dataset not only of local observations but also observations about which people were willing to share their knowledge to offer more transparent information about the protection status of our oceans. These observations can help reduce the number of 'paper parks' (i.e., MPAs that are legally designated, but not effective) and support climate-resilient 'blue economies.'

Establishing MPAs does not imply conservation success [33]. The creation of an MPA can be described in official documents, but this does not constitute a *de facto* MPA, i.e., a protected area with effective management and enforcement in place. MPAs that only exist on paper are 'paper parks', and do not generate any actual conservation benefits; in fact, they contribute to an unjustified sense of achievement [33]. For example, European MPAs allow for the most destructive form of fishing, bottom trawling, which implies a failure in their stated goal of protecting marine biodiversity [34]. Indeed, European MPAs suffer from an even greater trawling intensity than the surrounding areas [35].

The responses gathered can be used to differentiate 'paper parks' from the *de facto* MPAs. If most of the respondents from the different stakeholder groups replied with 'moderate' or 'very intense fishing' in a no-take MPA, where no fishing should occur, we may have identified a potential 'paper park'. In contrast, if most of the respondents from the different groups replied with 'light fishing' or 'no fishing' in a no-take MPA, the confidence in having a *de facto* MPA was quite high. This method would be the very first step to identifying potential 'paper parks,' which then would require further research to confirm the conclusions drawn from the stakeholders' statements. Going forward, an indicator system that points to either 'paper parks' or functional MPAs could be developed. However, more data and analysis are necessary to develop such an indicator, which we aim to undertake in future studies.

In the current research, we believe that one of the reasons for the overall high response rate was due to the personalized approach in contacting people and the seemingly 'simple' question. Respondents were deliberately not provided with a descriptive guide to the categories, as we chose to emulate the straightforward categorization of service satisfaction available at some airports (e.g., at Heathrow, near London, UK), where one can express an overall satisfaction (or dissatisfaction) without specifying what aspect of 'service' was evaluated, or how the aspect was assessed. Thus, the one-question questionnaire was also designed such that everyone could understand, regardless of origin or background, i.e., 1 = no fishing; 2 = light fishing; 3 = moderate fishing; 4 = very intense fishing. This approach probably was the reason for the high response rate of over 40%, and paradoxically the reason why many respondents answered by sending additional information and/or documents on the MPA in question.

In the dataset, we also entered in each record whether the responses included additional information. Extra information included any documents/links, a brief explanation for their selection, whether they provided any extra answers, and whether they facilitated

the name of another contact to reach out to. Among fishers and journalists, the responses—if any—tended not to include information beyond the MPA fishing level, which may reflect either a lack of time, interest, or trust about the effectiveness of the participative process [36]. In marine spatial planning and other conservation initiatives and tools, participatory approaches are indispensable for an inclusive MPA governance and decision-making that consider people's values, preferences, and interactions to minimize conflicts [36,37].

Community involvement in actual biodiversity protection can be mainly achieved through bottom-up governance approaches. Bottom-up governance empowers the community by involving it directly, while top-down governance is how governments regularly function [38]. A locally appropriate balance of top-down and bottom-up governance can lead to effective MPA planning [39,40]. Considering the empowerment of locals and the indigenous communities is key for conservation initiatives, but especially important in bottom-up governance as it offers the opportunity to rely on their local knowledge for the planning, implementation, and management of MPAs [41]. This type of governance can also help minimize the deleterious effects of ill-defined goals and definitions that prevent effective implementation and enforcement [34].

The most common extra information provided by the recipients in this study were peer-reviewed articles, newspaper articles, and reports, which were presented along with their own perceptions of the MPA. The perceptions were often categorized as anecdotal, but local stakeholder perceptions provide unique interpretations and insights that can contribute to a positive or negative evaluation of local environmental initiatives [42].

## 4. Conclusions

When coupled with poor governance, issues of limited funds, absent socio-economic incentives, limited community involvement [33], and the urgency to deliver on global conservation targets contribute to an increasing number of 'paper parks'. We presented a simple, affordable, low-tech, systematic, and replicable methodology to gather information based on personalized emails, using a one-question questionnaire, which could help to distinguish 'paper parks' from *de facto* MPAs. The method presented is viable for gathering information from different stakeholders, of which many have a close relationship with an MPA due to their expertise, job, and network. There were differences in the response rates among the different stakeholder groups, and further analysis would determine the extent to which the perceptions of an MPA's protection status vary by group. Of those that replied to the personalized email, 66% provided us with the fishing level of the MPA that we asked for.

The way and the degree to which different stakeholders shared their knowledge is not only of vital importance to the current study, but also vital to future knowledge co-production, policymaking, and monitoring. Knowledge co-production is the mechanism of bridging the gap between knowledge and action, where the "*research is conducted collaboratively, inclusively, and in a respectful and engaged manner [ . . . ] with the idea of creating actionable science and benefits to the partners involved*" [43]. Therefore, *ex situ* methods to gather information collaboratively (such as our one-question questionnaire) are important for the co-production of knowledge on the effectiveness of solutions for environmental issues such as in MPAs. Furthermore, we intend to expand our database to add local knowledge to more MPAs per country. We also plan to maintain the links to MPAtlas, Protected Planet (WDPA), FishBase, and SeaLifeBase in the future, to prevent the accumulation of non-functional links.

The success of MPAs depends on multi-stakeholder involvement during different stages of an MPA [23] and the dissemination of solutions-oriented knowledge about MPAs [44]. Moreover, an equal and respectful co-production of knowledge about marine ecosystems will improve participation in MPA management at a global level. Adding non-conventional knowledge to other initiatives that are mainly based on literature reviews, as in the case of so-called 'blue corridors' [45], will improve the connectivity of the habitats used by the exploited species, and ultimately improve the prospects of food security.

The Sea Around Us' research on MPAs and catch data serves as a complement to the data available on MPAs, providing a more thorough understanding of oceans and fisheries by considering different stakeholders' knowledge, which will continue to inform managers and policy, leading to a greater protection of marine ecosystems and biodiversity.

MPAs cannot be seen as a tool that can be used without considering the ecological, political, and social contexts within which they and their respective communities are embedded [46,47]. Considering those contexts is key to better incorporate community objectives and knowledge in the planning of MPAs [47]. Acceptance of an MPA, which is an outcome of stakeholder participation and community support, is vital for its successful management, enforcement, and monitoring [31,39]. It can have substantial positive impacts, such as contributing to an increase in hard coral cover in tropical areas [48]. However, attitudes, interactions, and perceptions can depend on the culture of the people involved [49], and the perceptions can be different even within the same type of stakeholders [50,51]. This is precisely why it is important to find a method of gathering and monitoring these stakeholder perceptions, as the methodology here presented.

Sending personalized emails with only one question is a method that appropriately elicits answers for the conservation status of MPAs from various stakeholder groups. The conciseness of the question motivated many respondents to provide additional data that were extremely helpful to the research. This information advanced the general knowledge about the status of MPAs on a global scale by equitably considering locals.

Various factors affect the performance and success of MPAs, such as political commitments and financial support [52,53], or the characteristics and location of the MPA itself [54]. However, a knowledge of local stakeholder perceptions is key to assessing the combined effect of these factors on the actual protection that MPAs provide to local biodiversity. This research presents the very first step to identifying potential 'paper parks', which now requires further research to confirm or deny the conclusions drawn from the stakeholders' statements. This is a first step for further investigations and subsequent research that focus on overcoming 'paper parks', key to properly assessing the *de facto* protection status of our oceans.

**Author Contributions:** V.R. and D.P. conceptualized the research idea of emailing various stakeholders around the world about their perception of the fishing level within select MPAs. V.R. developed and improved the scientific methodology. V.R. and T.M. collected the data from stakeholders, organizing it in a database style. Once the data acquisition phase was complete, T.M. compiled the research outcomes and data into webpage sections. S.O. uploaded and transferred the information onto the Sea Around Us website and database. V.R. and D.P. supervised all the steps involved. V.R. and T.M. wrote the first draft, which D.P. edited. All authors have read and agreed to the published version of the manuscript.

**Funding:** This is a product of Sea Around Us, based at the University of British Columbia, which is supported by the Oak Foundation, David and Lucille Packard Foundation, Marisla Foundation, Minderoo Foundation, Paul M. Angell Family Foundation, MAVA Foundation, Oceana, RARE, and the Summit Foundation. The corresponding author, Veronica Relano, received the support of a fellowship from "la Caixa" Foundation (ID 100010434). The fellowship code is LCF/BQ/AA18/11680035.

**Institutional Review Board Statement:** This study was performed under the approval of UBC's Office of Research Ethics (# H19-03029, 29 November 2019) for studies involving humans.

**Informed Consent Statement:** Informed consent was obtained from all subjects involved in the study.

**Data Availability Statement:** The MPA data produced by this research is directly accessible on Sea Around Us under the 'Tools & Data' tab on the toolbar (http://www.seaaroundus.org/data/#/mpa (accessed on 10 May 2022)).

**Acknowledgments:** We thank the many colleagues who have responded to our questionnaire about the effectiveness of their respective country's MPAs. A special thank you to Shannon Edie from the UBC Department of Statistics for helping us with the big dataset, identifying bugs, and providing insightful and technical advice.



**Conflicts of Interest:** The authors declare no conflict of interest. The funders had no role in: the design of the study; in the collection, analyses, or interpretation of data; in the writing of the manuscript; and in the decision to publish the results.

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
