# Peer review of "Stakeholder Perceptions Can Distinguish ‘Paper Parks’ from Marine Protected Areas"

_sustainability, doi:10.3390/su14159655_

Round 1
Reviewer 1 Report
· There is not a clear objective established. In the last paragraph of Introduction, authors mention: “Here we present how the information was collected and demonstrate how stakeholders from all over the world have contributed to sharing their knowledge towards a better understanding of marine conservation status through a simple, systematic, and easily replicable methodology”. This is not a clear objective. What do authors wanted to compare or highlight with this? I suggest a better objective.
· Title is confusing. I suggest Stakeholder perceptions helped distinguish “paper parks” from marine protected areas.
· In the abstract, I suggest the following edit in the line 14: In order to address this gap we recorded responses of stakeholders worldwide based on a questionnaire survey and added information in the Sea Around Us internet site.
· I recommend authors to include in the abstract what were the key findings based on the stakeholder’s observations recorded. Otherwise, this work it appears just to be an announcement of an internet site of the Sea around Us project.
· I urge authors to separate Results from Discussion. Otherwise, it is really complicated to visualize which were the results and which the discussion.
· The conclusion section does not have key findings of the work.
· In general, the work still needs to be clearly assembled, the objective needs to be clarified, the methodology explained better, results separate from the discussion and the conclusions clearly established.
Author Response
Thank you for your excellent comments. We have submitted a document responding to each revision.

Reviewer 2 Report
Dear authors!
Please check out the suggestions for improvements below.
Line 11: Move the fragment (NGOs, academics, Government civil servants, journalists, and fishermen) on lines 21-22 to line 11 and insert it after the word stakeholders.
Line 19: NGOs - the abbreviation decoding is necessary at the first mention in the Abstract.
Lines 22-23: Delete one additional word in the sentence.
Line 25: If you use the concept of ‘paper parks', explain it. In the Abstract and the text of a scientific article, it is better to avoid figurative phrases.
Line 31-34: The word biodiversity in the context of conservation is mentioned three times in sentences with a similar meaning. Change the sentences. Reduce the mention of biodiversity conservation to one and combine references 1 and 2 with common brackets.
Line 41: Move the fragment is a research initiative at the University of British Columbia that from line 55-56 and insert it after Sea Around Us on line 41. Consider splitting the resulting sentence (lines 41-44) into two shorter sentences.
Line 47: I propose to combine the paragraph on line 47 and the paragraph on line 52 into one paragraph.
Lines 48-49: I suggest deleting the fragment while providing additional links and information when available.
Line 52: I propose to delete the fragment based on data submitted annually by its member countries,
Line 81: Start the paragraph with a sentence that talks about the innovations of your work: (1) functional changes to the database; (2) questionnaire.
Line 81: EEZ – the abbreviation decoding is necessary at the first mention in the text of the article.
Line 82-84: I suggest deleting the fragment to include a link to ‘Marine Protected Areas', which opens a page with a brief narrative on that country's effort to protect its marine biodiversity, notably in the form of treaties or conventions that it is a member of
Line 84: I suggest deleting a fragment of Note that
Line 113: Combine paragraphs on lines 113, 117, 124 and 137 into one paragraph.
Line 113: delete the word different
Line 114: I suggest deleting the fragment (with a follow up in case of non-response)
Line 114: NGO – the abbreviation must be deciphered at the first mention in the text of the article.
Lines 115-116: I suggest deleting the fragment these were people that appeared to have close relationships with various MPas due to their expertise, profession or network
Line 117: I propose to delete the fragment sent out to stakeholders
Line 118: delete the word essentially
Lines 124-125: Move sentence This study was performed under the approval of UBC's Office of Research Ethics (#124 H19-03029, November 29, 2019) for studies involving humans. in the Institutional Review Board Statement
Lines 126-127: remove repetition to have a representation of each group of stakeholders (i.e. academics, fisher representatives, civil servants, journalists and NGO staff)
Lines 133-135: I suggest deleting the paragraph completely
Lines 140-141: delete For the analysis, in the case of ties, i.e. where multiple fishing levels had the same frequency for the same stakeholder group, the
Lines 154-155: link a sentence.
Line 159: delete it Whenever possible,
Lines 165-166: We plan to maintain the links to MPAtlas, Protected Planet (WDPA), FishBase, and SeaLifeBase in the future, such as to avoid dead links. - find a place and move this sentence to the end of the manuscript, for example, to line 280, where you will talk about the prospects for new changes to improve the efficiency of your work with respondents and gain new knowledge about the MRA.
Lines 169-170: Country-level MPA details show a map with approximate MPA location and biodiversity information. I propose to change the sentence, for example, А map with approximate MPA location and biodiversity information and сountry-level MPA details.
Line 195: correct the word recipients to the singular
Lines 220-223: I suggest removing Participation would have likely been higher if we had called them, but this was not done to maintain the same methods as used for the other stakeholder groups. Going forward, we suggest more follow-ups with fishers, using other communication channels. However,
Lines 223-226: I propose to combine with the upper paragraph overall the participation of all stakeholders generated a big dataset not only of local observations but of people that were willing to share their knowledge to offer more transparent information about the protection status of our oceans, reduce the number of ‘paper parks' and support climate-resilient blue economies.
Line 226: Blue economy – quotes are needed.
Line 256: I suggest moving the first paragraph in the Conclusion and inserting it before the last paragraph. I propose to combine the Results and discussion with the Conclusions in their current state. The Conclusions section should include two main conclusions arising from this work, each of which should consist of 1-2 sentences.
Author Response

(The authors gave the same response as above.)

Reviewer 3 Report
1. This manuscript intends to add a new feature of stakeholder perceptions of the fishing level (or fishing intensity) for each selected MPA to an established MPAs website. By doing so, it claims that the result can help distinguish ‘paper parks’ from MPAs and facilitate a better understanding of the de facto protection and conservation status of MPAs globally. It structurally and meticulously describes the steps and methods to do so and thus offers an overall view of the conservation status of MPAs globally. However, two points below need to be addressed in order to demonstrate the merits of ‘stakeholder perceptions’, as claimed by authors, in its contribution to a better understating of MPAs conservation status.
2. While the results show the fishing level of each MPA perceived by each stakeholder group (as seen figure 3), it is very likely that different stakeholder groups have different perceptions pertaining to the same MPA. In this case, how these differing data (perhaps from no fishing to very intense fishing) can be used to distinguish ‘paper parks’ from MPAs. It is noted that the title said that stakeholder perceptions can help fulfill this purpose. However, the manuscript said nothing on how to ‘integrate’ these differing perceptions into an indicator for the level of effectiveness of each MPA. It is suggested that the manuscript elaborates more on the relations between the data of fishing intensity derived from stakeholder groups and the effectiveness of MPAs and further on the situations that there might be imitations in referring this relation.
3. It is noted that the manuscript explains the implications for low response rate of fishers group compared to NGOs, Academics and Governments groups (as seen figure 4) and thus emphasizes the importance of participation of fishers group in improving MPAs’ performance. However, it is found that the response rate of journalists is much lower than the fishers group but the manuscript said nothing on this finding. Does this low response rate reflect any implications for the MPA governance and status or other aspects? It is suggested that the manuscript dwell on this point.
Author Response
Thank you very much for your time

Round 2
Reviewer 1 Report
After various review rounds, authors failed to establish a clear objective of the work. Also, authors did not follow my suggestion to split Results from Discussion which would better identify what authors did for that what they compare to. I regret I am no longer participate in another review.
Author Response
Dear reviewer,
Thank you for your time. We would like to clarify that there has been only one round of reviews. And now in the second one, we have incorporated even further information and details to explain the objective of our work. Please look at the completed answers we have submitted to the editors, there are more explanations, including the new paragraphs and information added.
Thank you for your patience and your time.
Reviewer 2 Report
Dear authors!
The article is far from ideal, but it has been improved now.
All the best
Author Response
Dear reviewer,
Thank you for your time and your words. We have incorporated more information and details to explain the objective of our work. Please look at the completed answers we have submitted to the editors, there are more explanations, including the new paragraphs and information added, which we hope that improves our paper further.
Thank you for your patience and your time.
This manuscript is a resubmission of an earlier submission. The following is a list of the peer review reports and author responses from that submission.
Round 1
Reviewer 1 Report
Dear authors, please read suggestions in the file "Suggestions.doc".

Reviewer 2 Report
Mak et al. provided an overview on the new addition of marine protected areas (MPA) information on the Sea Around Us database and website for all the countries’ Exclusive Economic Zones. A brief narrative was provided on the country’s effort to protect its marine biodiversity, and questionnaire responses on the perceived protection level of MPAs from over 4,000 stakeholders were also shown. Quotes and references were also provided for easy access and updates. The information may be useful for planning and management of existing MPAs to implement their protective features.
I deeply appreciate the efforts of authors and supporting team in compiling and presenting the MPA and fishery data to allow a better understanding of the conservation status of marine fisheries, and the actual effectiveness of MPA in this manner. However, for me, the manuscript is considered more like a kind of communication or conservation news rather than a research article that reports scientifically sound experiments and provides a substantial amount of new information.
In terms of a research article, in addition to the introduction of feature and technical function of the website, it is also suggested to include some preliminary results or discoveries which are worth to be addressed to the field managers and conservation scientists. I also expect to read more discussions on how the collected data can potentially contribute to the perceived information gaps on the planning and management of MPAs, especially in developing countries with generally weaker relevant knowledge, funding and enforcement to maintain an effective MPA.
For the questionnaire about perceived actual fishing levels in selected MPAs, I am not sure whether the respondents were previously provided a descriptive guide to help them assess in a more scientifically manner, since the available choices of “light fishing”, “moderate fishing” and “very intense fishing” can mean very different for the respondents. I understand that the collected responses were compiled into a collective result so that it can be visualized on the map, but the raw responses with descriptions can be equally important, particularly when there were apparent discrepancies among responses.
Reviewer 3 Report
- I fully realize the gigantic importance of the Sea Around Us Project has as repository database worldwide, and the great input this project has for the advance of fishery trends worldwide. However, this manuscript is basically presenting a technical report regarding a new add up to the database rather than presenting a traditional scientific contribution of a given situation.
- I strongly recommend authors to completely change the structure of the information in the manuscript and chiefly focus on the premise authors mentioned addressed as a scientific prospection, based on a questionnaire applied to stakeholders worldwide. So, in this sense, authors MUST establish an objective first at the very last sentence of the last paragraph of the Introduction section. Something like: “The aim of this work was to compile (describe or whatever is the objective) information from stakeholders associated to a marine protected area worldwide……..”
- Consequently, the title must be changed to something like: “Stakeholder perception of attributes in the Marine Protected Areas Worldwide ”
- Authors need to change the perspective of the abstract accordingly to my suggestions. I mean, it must be a scientific contribution attempting to portrait the perception of stakeholders worldwide, and their opinion of the MPAs, and link these results to the Sea Around Us database to better integrate the current information. Otherwise, the current abstract is way to technical report of some sort.
- Table 1 is awkward. What did authors want to show with this? Is this just the template authors used to compile the information? The information requested in the questionnaire must be presented in a different manner, such a graph, but with collected information!
- When authors mention, in Material and methods, that “Whenever feasible MPAs were hyperlinked to MPAAtlas and Protected Planet? What did authors mean with this hyperlink? Which type of data were collected and integrated?
- I strongly suggest to splitting the section Results and Discussion in order to better visualize which were the results of the work and how these are compared with other information including quotation.
- Figure 1. What is the new information obtained through this work and depicted in Figure 1? Is there any possibility to highlight that?
- Figure 2. What new information authors obtained through the survey to stakeholder? Which can be depicted here? It is hard to understand.
- Figure 3. The Same happens with this figure. Where is the new information or how the information was used to assemble the add-up?
- It is the very first time in more than 30 years of reviewing papers that I see a very short Results and Discussion section. On one hand, it is hard to visualize the results and on the other there is not any comparison (Discussion) based on quotes from somebody else.
- The Conclusions section is difficult to understand because the key findings based on the survey are not presented.
- I strongly recommend authors to better assemble the manuscript and include a more detailed methodology on how they collected information from stakeholders, and how this information was integrated, analyzed, and compared by the authors. Otherwise, this manuscript is a technical report.
- Unfortunately, at its present form the manuscript is rejected.